# A virus-encoded protein suppresses methylation of the viral genome through its interaction with AGO4 in the Cajal body

Liping Wang[1,2], Yi Ding[1,2], Li He[1], Guiping Zhang[1], Jian-Kang Zhu[1], Rosa Lozano-Duran[1]*

[1]Shanghai Center for Plant Stress Biology, Center for Excellence in Molecular Plant Sciences, Chinese Academy of Sciences, Beijing, China; [2]University of the Chinese Academy of Sciences, Beijing, China

**Abstract** In plants, establishment of de novo DNA methylation is regulated by the RNA-directed DNA methylation (RdDM) pathway. RdDM machinery is known to concentrate in the Cajal body, but the biological significance of this localization has remained elusive. Here, we show that the antiviral methylation of the *Tomato yellow leaf curl virus* (TYLCV) genome requires the Cajal body in *Nicotiana benthamiana* cells. Methylation of the viral genome is countered by a virus-encoded protein, V2, which interacts with the central RdDM component AGO4, interfering with its binding to the viral DNA; Cajal body localization of the V2-AGO4 interaction is necessary for the viral protein to exert this function. Taken together, our results draw a long sought-after functional connection between RdDM, the Cajal body, and antiviral DNA methylation, paving the way for a deeper understanding of DNA methylation and antiviral defences in plants.

*For correspondence:
lozano-duran@sibs.ac.cn

Competing interests: The authors declare that no competing interests exist.

## Introduction

DNA methylation in cytosine residues is a conserved epigenetic mark essential for protecting the eukaryotic genome against invading nucleic acids, namely viruses and transposable elements. In plants, establishment of de novo DNA methylation is believed to be regulated by the RNA-directed DNA methylation (RdDM) pathway. The canonical RdDM pathway requires the activity of two plant-specific RNA polymerase II-related enzymes, Pol IV and Pol V, and leads to cytosine methylation in a sequence-specific manner. In brief, the current understanding of RdDM is as follows: Pol IV generates RNA transcripts subsequently converted to double-stranded RNA (dsRNA) by RDR2 (*Haag et al., 2012*; *Law et al., 2011*), and then diced into 24-nt siRNAs by DCL3 (*Xie et al., 2004*); the resulting 24-nt siRNAs are loaded into AGO4 (*Zilberman et al., 2003*), which is guided to scaffold RNA molecules generated by Pol V via sequence complementarity and recruits the de novo methyltransferase DRM2 (*Böhmdorfer et al., 2014*; *Chan et al., 2005*; *Gao et al., 2010*; *Zhong et al., 2014*), which in turn catalyses methylation of adjacent DNA sequences. RdDM generally creates a chromatin environment refractive to gene expression. Part of the RdDM machinery, including AGO4, was found to concentrate in the Cajal body, a subnuclear compartment that is the site of maturation of ribonucleoprotein complexes (*Li et al., 2006*). This observation led to the suggestion that Cajal bodies might be a center for the assembly of an AGO4/NRPE1/siRNA complex, which would facilitate RdDM at target loci (*Li et al., 2006*). However, the biological relevance of the Cajal body localization of RdDM components, and of this compartment for DNA methylation, has so far not been demonstrated.

Geminiviruses are a family of plant viruses with circular single-stranded (ss) DNA genomes infecting multiple crops and causing dramatic yield losses worldwide. The geminiviral genome replicates in the nucleus of the infected cell by using the host DNA replication machinery, made available following the viral re-programming of the cell cycle (reviewed in *Hanley-Bowdoin et al., 2013*). During viral multiplication, the ssDNA genome generates a double-stranded (ds) DNA intermediate, which then undergoes rolling-circle replication and recombination-dependent replication (reviewed in *Hanley-Bowdoin et al., 2013*). Nevertheless, the cellular and molecular details underlying these essential initial steps of the viral infection cycle, including the subnuclear localization of viral ss and ds DNA accumulation and of the processes leading to their production, are to date mostly unknown.

Notably, the geminiviral genome forms minichromosomes and is subjected to epigenetic modifications, including cytosine DNA methylation and histone modifications (*Ceniceros-Ojeda et al., 2016*; *Deuschle et al., 2016*; *Jackel et al., 2016*; *Kushwaha et al., 2017*; *Raja et al., 2008*; *Wang et al., 2018*). The findings that methylation of viral DNA negatively impacts viral replication (*Brough et al., 1992*; *Ermak et al., 1993*), that different geminivirus-encoded proteins have evolved to suppress DNA methylation (*Buchmann et al., 2009*; *Ismayil et al., 2018*; *Raja et al., 2008*; *Rodríguez-Negrete et al., 2013*; *Tu et al., 2017*; *Wang et al., 2018*; *Yang et al., 2013*; *Yang et al., 2011*; *Zhang et al., 2011*), and that methylation of the viral genome correlates with host resistance or recovery (*Butterbach et al., 2014*; *Ceniceros-Ojeda et al., 2016*; *Raja et al., 2008*; *Torchetti et al., 2016*; *Yadav and Chattopadhyay, 2011*) strongly support the idea that the plant-mediated methylation of the viral DNA acts as an antiviral defence mechanism, underscoring DNA methylation as an active battlefield in the interaction between plants and geminiviruses.

During the infection by the geminivirus *Tomato yellow leaf curl virus* (TYLCV), the essential virus-encoded protein V2 has been shown to suppress DNA methylation (*Wang et al., 2014*). V2 interacts with the histone deacetylase HDA6 in *Nicotiana benthamiana*, competing with the recruitment of the maintenance CG methyltransferase MET1 (*Woo et al., 2008*) and ultimately reducing viral DNA methylation (*Wang et al., 2018*). Nevertheless, silencing of *HDA6* results in limited complementation of a V2 null mutation in the virus and only a partial reduction in viral DNA methylation (*Wang et al., 2018*), suggesting that V2 might counter methylation through additional interactions with host factors.

In this work, we show that V2 from TYLCV interacts with the central RdDM component AGO4; that AGO4 plays a role in the defence against TYLCV; and that V2 interferes with the AGO4 binding to the viral DNA. Importantly, our results indicate that the viral DNA gets methylated in a Cajal body-dependent manner in the absence of V2, since abolishment of Cajal body formation by silencing of the gene encoding its signature component coilin drastically reduces methylation of the viral genome. This idea is further supported by the finding that the activity of V2 as a suppressor of viral DNA methylation requires Cajal body localization of the V2-AGO4 interaction. In summary, our results draw a functional connection between antiviral RdDM and the Cajal body in plants, and illustrate how TYLCV has evolved a protein, V2, to counter this plant defence mechanism.

## Results

### V2 from TYLCV interacts with AGO4 from *Nicotiana benthamiana* and tomato

With the aim of gaining insight into the functions of V2 from TYLCV in the plant cell, we used transient expression of GFP-tagged V2 in infected leaf patches of *N. benthamiana* followed by affinity purification and mass spectrometry (AP-MS) to identify plant interactors of this viral protein in the context of the infection (*Wang et al., 2017a*). Interestingly, we identified the two AGO4 paralogs in *N. benthamiana* (NbAGO4-1 and NbAGO4-2) as associated to V2-GFP (*Figure 1A*; *Wang et al., 2017a*); these interactions were confirmed by co-immunoprecipitation (co-IP) and split-luciferase assays (*Figure 1B,C*).

There are four *AGO4* orthologues in tomato (*SlAGO4a-d*) (*Bai et al., 2012*), the natural and economically relevant host of TYLCV (*Figure 2A,B*). All four SlAGO4-encoding genes are expressed in basal conditions in tomato leaves, although *SlAGO4c* and *d* show low expression levels; *SlAGO4b*, *c*, and *d* are slightly upregulated by TYLCV infection (*Figure 2—figure supplement 1*). *SlAGO4a*, *SlAGO4b*, and *SlAGO4d* were cloned and the encoded proteins confirmed as interactors of V2 in

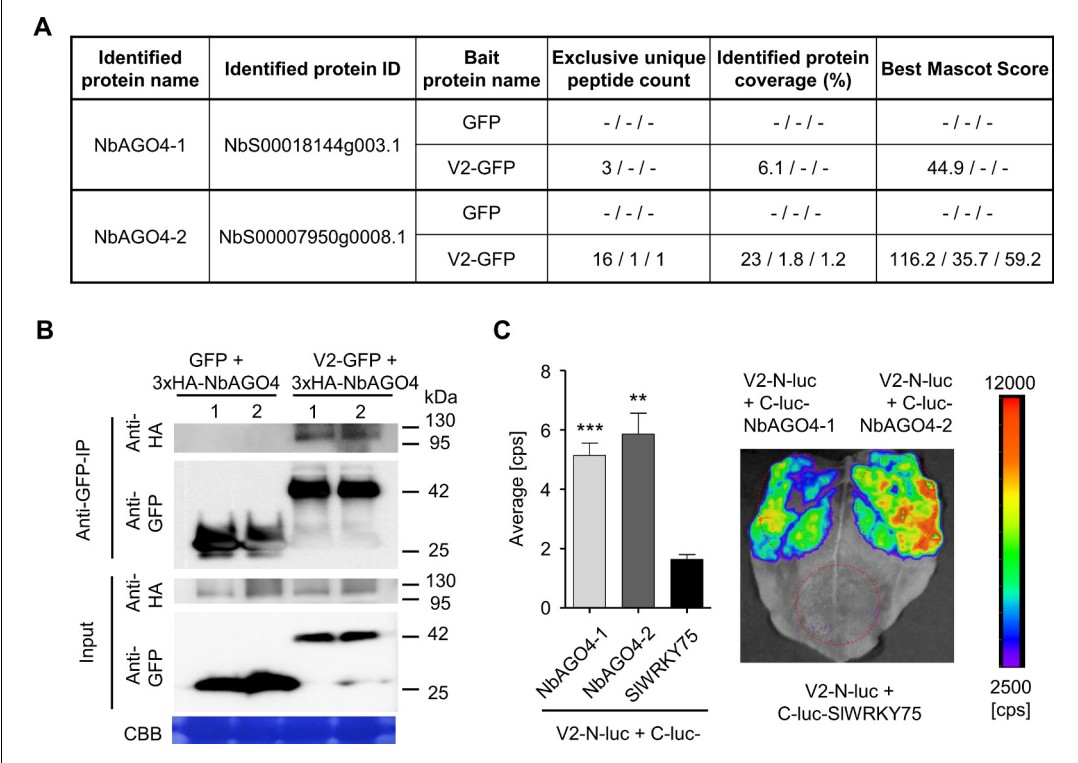

**Figure 1.** V2 interacts with AGO4 from *N. benthamiana*. (**A**) Unique peptide count, protein coverage, and best Mascot Score of NbAGO4-1 and NbAGO4-2 co-immunoprecipitated with V2-GFP, as identified by affinity purification followed by mass spectrometry (AP-MS). Results from three independent biological repeats are shown. "-" indicates no peptide was detected. (**B**) 3xHA-NbAGO4-1 and 3xHA-NbAGO4-2 specifically interact with V2-GFP in co-immunoprecipitation (co-IP) assays upon transient expression in *N. benthamiana*. Free GFP was used as negative control. CBB, Coomassie brilliant blue staining. Three independent biological replicates were performed with similar results. (**C**) NbAGO4-1 and NbAGO4-2 interact with V2 in split-luciferase assays. V2-N-luc and C-luc-NbAGO4-1/2 were transiently co-expressed in *N. benthamiana*; C-luc-SlWRKY75 is used as negative control. The luciferase bioluminescence from at least three independent leaves per experiment was imaged 2 days after infiltration. The average bioluminescence, measured in counts per second (cps), as well as an image of a representative leaf are shown. Values represent the mean of three independent biological replicates; error bars indicate SEM. Asterisks indicate a statistically significant difference (according to Student's *t*-test, **: p<0.01, ***: p<0.001) compared to the negative control.

co-IP and split-luciferase assays (*Figure 2C,D*). Our results therefore show that V2 interacts with AGO4 in two host species, tomato and *N. benthamiana*.

## V2 counters the AGO4-dependent methylation of the viral genome to promote virulence

In order to evaluate the contribution of V2 to the viral infection, we generated an infectious TYLCV clone carrying a G-to-A mutation in the fifth nucleotide of the V2 open reading frame (ORF), which converts the second codon (encoding tryptophan) to a stop codon (*Figure 2—figure supplement 2*), making it unable to produce the V2 protein (TYLCV-V2null). In agreement with previous results (*Wartig et al., 1997*), V2 is required for full infectivity in both tomato and *N. benthamiana*, since the V2 null mutant accumulates to very low levels and produces no noticeable symptoms in systemic infections (*Figure 2—figure supplement 2*).

Next, we sought out to test whether knock-down of *AGO4* could partially complement the lack of V2 during the TYLCV infection. For this purpose, we employed virus-induced gene silencing (VIGS) to silence both *NbAGO4-1* and *NbAGO4-2*. VIGS efficiently knocked-down both *NbAGO4* orthologues, but did not affect accumulation of the transcript of the close homologue *NbAGO6* (*Figure 3A*); *AGO4*-silenced plants did not display any obvious developmental abnormalities (*Figure 3B*). Expression of *NbAGO4-1* or *NbAGO4-2* was not affected by TYLCV infection, neither in silenced nor in non-silenced plants (*Figure 3C,D*). Mutation in V2 does not affect viral replication

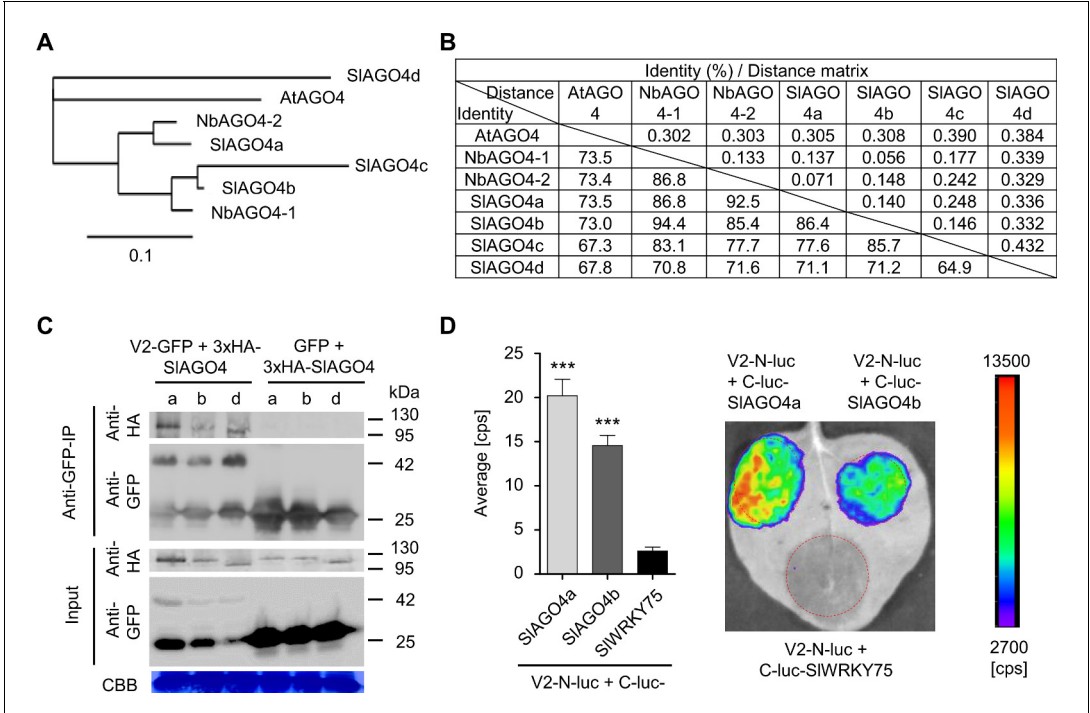

**Figure 2.** V2 interacts with AGO4 from tomato. (**A**) Phylogenetic tree of AtAGO4, NbAGO4, and SlAGO4 proteins. The phylogenetic analysis was performed with phylogeny.fr (**Dereeper et al., 2010**; **Dereeper et al., 2008**). (**B**) Pairwise identity and genetic distance matrix among AtAGO4, NbAGO4 and SlAGO4 proteins. The analysis was performed by Geneious (https://www.geneious.com). (**C**) 3xHA-SlAGO4a, 3xHA-SlAGO4b, and 3xHA-SlAGO4d specifically interact with V2-GFP in co-immunoprecipitation (co-IP) assays upon transient expression in *N. benthamiana*. Free GFP was used as negative control. CBB, Coomassie brilliant blue staining. Three independent biological replicates were performed with similar results. (**D**) SlAGO4a and SlAGO4b interact with V2 in split-luciferase assays. V2-N-luc and C-luc-SlAGO4a/b were transiently co-expressed in *N. benthamiana*; C-luc-SlWRKY75 is used as negative control. The luciferase bioluminescence from at least three independent leaves per experiment was imaged two days after infiltration. The average bioluminescence, measured in counts per second (cps), as well as an image of a representative leaf are shown. Values represent the mean of three independent biological replicates; error bars indicate SEM. Asterisks indicate a statistically significant difference (according to Student's *t*-test, ***: p<0.001) compared to the negative control.

The online version of this article includes the following figure supplement(s) for figure 2:

**Figure supplement 1.** *SlAGO4* expression in TYLCV-infected and control tomato plants.
**Figure supplement 2.** V2 is essential for systemic TYLCV infection in tomato and *N. benthamiana*.

(**Wartig et al., 1997**), and therefore viral accumulation in local infections in *N. benthamiana* (leaf patch agroinfiltration assays; see **Figure 3—figure supplement 1**) was not different between the wild-type virus and the V2 null mutant virus (**Figure 3E**); in both cases, *AGO4* silencing led to a not statistically significant but reproducible trend to higher viral accumulation, suggesting an antiviral role for AGO4 (**Figure 3E**). Interestingly, in systemic infections (see **Figure 3—figure supplement 1**), *AGO4* silencing mildly increased viral accumulation of the wild-type TYLCV (1.33-fold), but dramatically improved performance of the V2 null mutant virus (3.23-fold), suggesting that one of the main roles of V2 during the viral infection is the suppression of AGO4 function (**Figure 3F**).

In light of the role of AGO4 in RdDM and to directly assess the impact of V2 and AGO4 on the methyl-state of the viral DNA, we used bisulfite sequencing (BS-seq) to measure DNA methylation of the intergenic region (IR) of the viral genome, which presents the highest methylation levels during the infection (**Piedra-Aguilera et al., 2019**). As shown in **Figure 4A**, cytosine methylation in this region in all contexts (CG, CHG, and CHH) was almost undetectable in the wild-type viral genome in local infections at 3 or 9 days post-inoculation (dpi), while it reached ~60% and~80%, respectively, in the V2 null mutant (**Figure 4A**; **Figure 4—figure supplement 1A**; **Supplementary file 1**). These results indicate that V2 can prevent or revert methylation of the viral genome during the infection, which occurs rapidly in the absence of this protein.

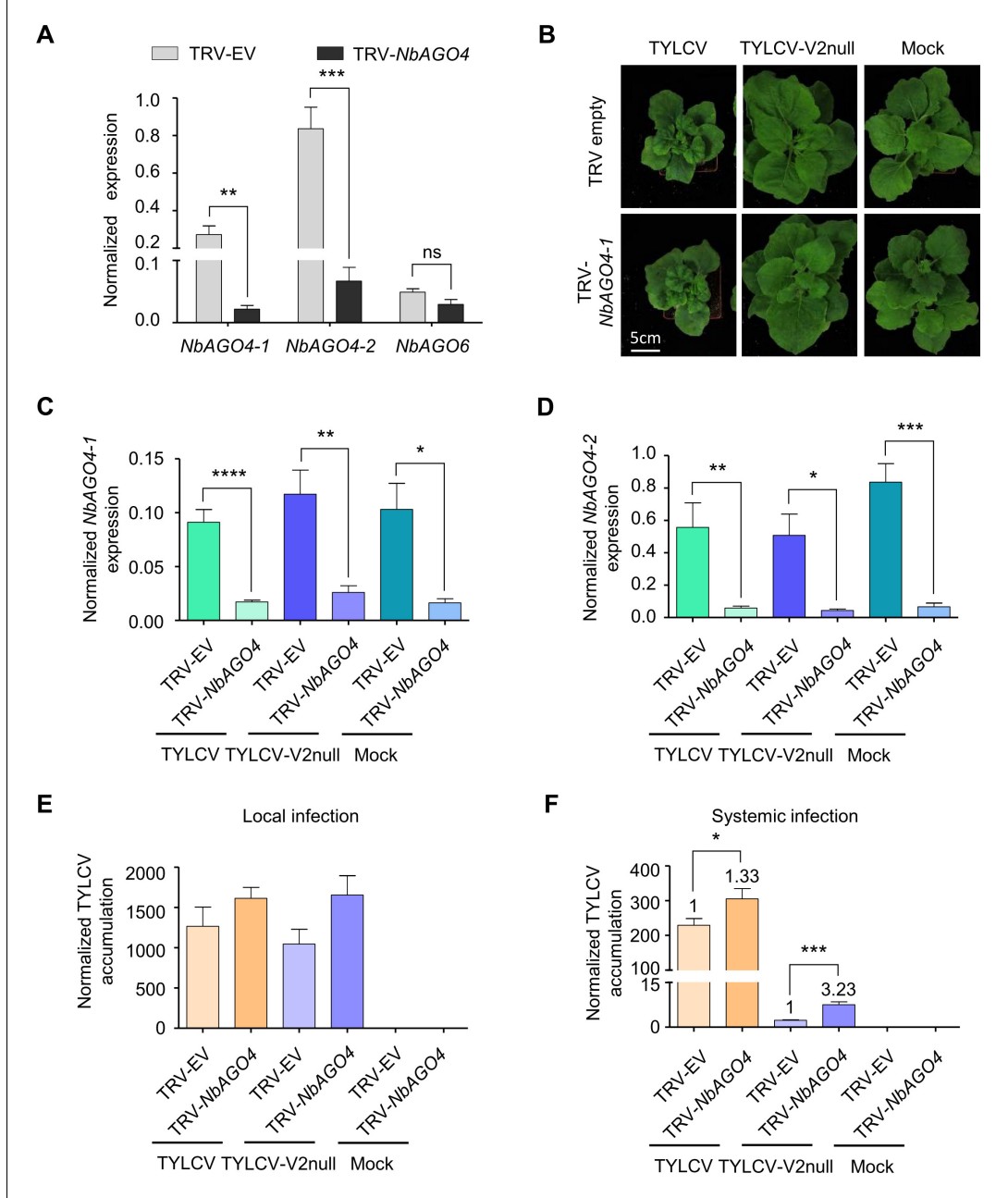

**Figure 3.** V2 counters the AGO4-dependent antiviral defence to promote virulence. (A) Expression of *NbAGO4-1*, *NbAGO4-2,* and *NbAGO6* in *N. benthamiana* plants infected with TRV-EV (empty vector) or TRV-*NbAGO4*, measured by reverse transcription quantitative real-time PCR (RT-qPCR). Gene expression was normalized to *NbTubulin*. Values are the mean of four independent biological replicates; error bars indicate SEM. Asterisks indicate a statistically significant difference according to Student's *t*-test. **: p<0.01, ***: p<0.001, ns: not significant. (B) Representative pictures of *N. benthamiana* plants infected with the indicated combinations of viruses. Photographs were taken at 3 weeks post-inoculation (wpi). (C) *NbAGO4-1* expression in *NbAGO4*-silenced plants and control plants infected with TYLCV, TYLCV-V2null, or mock-inoculated at 3 wpi measured by RT-qPCR. Gene expression was normalized to *NbTubulin*. Values are the mean of six independent biological replicates; error bars indicate SEM. Asterisks indicate a statistically significant difference according to Student's *t*-test. *: p<0.05, **: p<0.01, ****: p<0.0001. (D) *NbAGO4-2* expression in *NbAGO4*-silenced plants and control plants infected by TYLCV, TYLCV-V2null, or mock-inoculated at 3 wpi measured by RT-qPCR. Gene expression was normalized to *NbTubulin*. Values are the mean of six independent biological replicates; error bars indicate SEM. Asterisks indicate a statistically significant difference according to Student's *t*-test. *: p<0.05, **: p<0.01, ***: p<0.001. (E) Viral (TYLCV) accumulation in local infections in *NbAGO4*-silenced or control plants, measured by qPCR. Infiltrated leaf patches from different plants were collected at 4 dpi. The experimental design is shown in *Figure 3—figure supplement 1A*. The accumulation of viral DNA is normalized to the *25S ribosomal RNA interspacer (ITS)*. Values are the mean of eight independent biological replicates; error bars indicate SEM. (F) Viral (TYLCV) accumulation in systemic infections in *NbAGO4*-silenced or control

*Figure 3 continued on next page*

Figure 3 continued

plants, measured by qPCR. Apical leaves from six plants were collected at 3 wpi. The experimental design is shown in *Figure 3—figure supplement 1B*. The accumulation of viral DNA is normalized to the *25S ribosomal RNA interspacer (ITS)*. Four independent biological replicates were performed with similar results; one representative result is shown. Values are the mean of six independent biological replicates; error bars indicate SEM. Asterisks indicate a statistically significant difference according to Student's *t*-test. *: p<0.05, ***: p<0.001. The relative fold change of viral accumulation between *NbAGO4*-silenced plants and control plants is shown above each column.

The online version of this article includes the following figure supplement(s) for figure 3:

**Figure supplement 1.** Experimental design for local and systemic TYLCV infection assays in *NbAGO4*- or *Nbcoilin*-silenced *N. benthamiana* plants.

We then compared the percentage of cytosine methylation in the intergenic region in local infections with the V2 null mutant TYLCV in basal conditions or upon *AGO4* silencing. Strikingly, *AGO4* silencing resulted in a ~30% decrease in the percentage of methylated cytosines in all contexts (*Figure 4B*; *Figure 4—figure supplement 1B*; *Supplementary file 1*), indicating that knock-down of *AGO4* can partially complement the lack of V2 at the level of the viral methylation state. This complementation suggests that (a) methylation of the viral DNA at least partially depends on AGO4 function and (b) V2 can at least partially counter AGO4-dependent methylation of the viral DNA.

As opposed to short-timed local infections, in systemic infections in *N. benthamiana*, which require longer timespans and involve viral cell-to-cell and long-distance movement, methylation of the wild-type viral genome could be detected at ~8–23% in all contexts (*Figure 4C*; *Figure 4—figure supplement 2*; *Supplementary file 1*). Interestingly, the methylation level tends to decrease upon *AGO4* silencing; this reduction (~12–31%) brings the methylation of the V2 null mutant genome back to wild type-like levels, again supporting the idea that AGO4-dependent methylation of the viral genome occurs during the infection and is partially countered by V2. Notably, the detected decrease in methylation correlates with the enhanced viral accumulation in the *AGO4*-silenced plants (*Figure 3F*).

## V2 does not hamper production or loading of vsiRNA but interferes with AGO4 binding to the viral RNA and to the viral genome

The canonical function of AGO4 in the RdDM pathway requires loading of siRNA and association to Pol V/the Pol V-dependent scaffold RNA, and results in the recruitment of DRM2 to the target loci and the subsequent methylation of the adjacent DNA (*Matzke et al., 2015*; *Matzke and Mosher, 2014*). Through physical interaction, V2 could affect AGO4 function on the viral genome in different ways, for example by impairing loading of viral siRNA (vsiRNA) onto this protein or by displacing endogenous interactors, such as Pol V or DRM2. In order to shed light on the molecular mechanism underlying the V2-mediated interference of AGO4-dependent methylation of the viral genome, we tested binding of AGO4 to the viral DNA in the presence or absence of V2 in local infections with TYLCV wild-type and the V2 null mutant, respectively, by Chromatin immunoprecipitation (ChIP). As shown in *Figure 5A*, 3xFLAG-NbAGO4-1 could bind both the IR and the V2-encoding region of the viral genome in the absence of V2 (TYLCV-V2null), but the signal decreased to background levels when V2 was present (TYLCV). Therefore, AGO4 has the capacity to bind the viral DNA molecule, but this binding is impaired by the virus-encoded V2 protein. AGO4 binding in the TYLCV V2 null mutant hence correlates with the detected increase in viral DNA methylation (*Figure 4A*).

Binding of AGO4 to the Pol V subunit NRPE1 has been previously detected and proposed to contribute to its recruitment to the target DNA (*Li et al., 2006*); however, we did not detect quantitative differences in the association of 3xFLAG-NbAGO4-1 with NRPE1 in the presence or absence of V2 in the context of the infection by AP-MS (*Supplementary file 2*). Therefore, we next evaluated whether V2 interferes with the binding of AGO4 to the viral RNA that could act as scaffold. For this purpose, we performed RNA immunoprecipitation (RIP) using 3xFLAG-NbAGO4-1 in locally infected samples; our results indicate that the presence of V2 in the wild-type virus causes a decrease in the binding of AGO4 to RNA molecules derived from the non-coding intergenic region (IR) and the adjacent V2 ORF (*Figure 5B*; *Figure 5—figure supplement 1*).

Several viral silencing suppressors encoded by different viruses have been shown to inhibit formation of AGO/sRNA complexes (e.g. *Burgyán and Havelda, 2011*; *Rawlings et al., 2011*; *Schott et al., 2012*). To test whether this strategy is also employed by V2, we immunoprecipitated

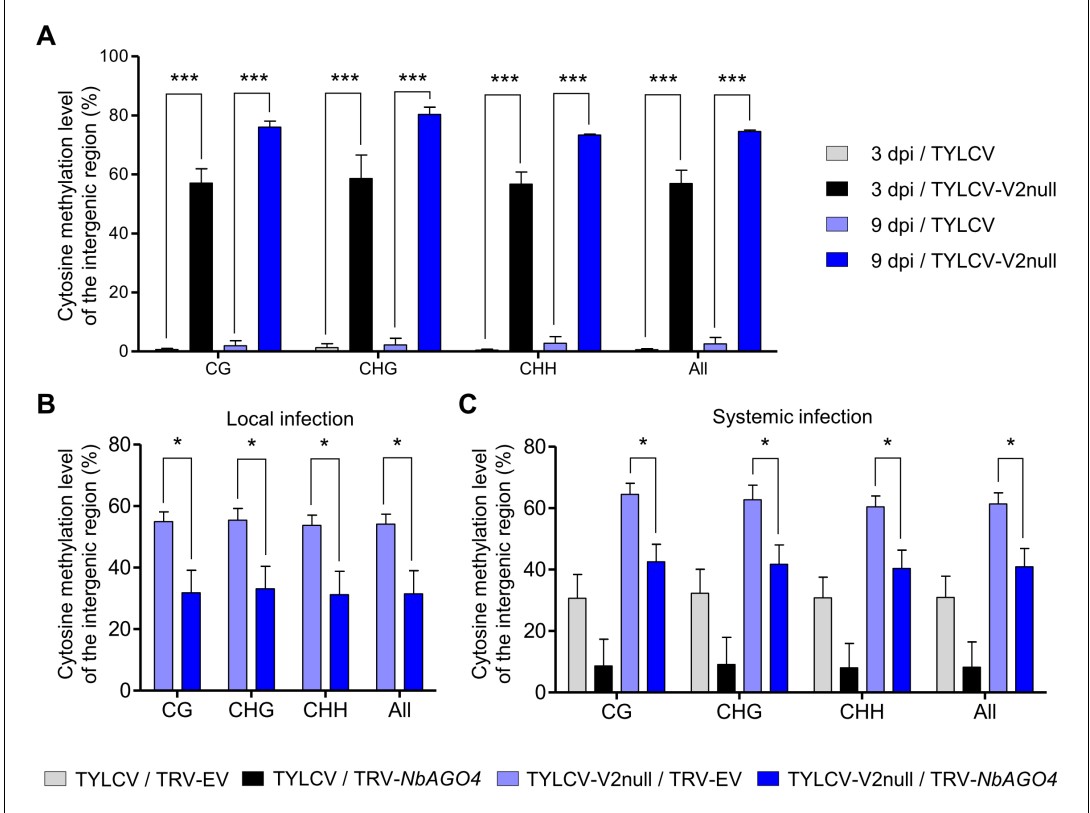

**Figure 4.** V2 suppresses the AGO4-dependent methylation of viral DNA. (**A**) Percentage of methylated cytosines in the intergenic region (IR) of TYLCV in local infection assays with TYLCV wild-type or V2 null mutant (TYLCV-V2null) in *N. benthamiana* at 3 or 9 days post-inoculation (dpi), as detected by bisulfite sequencing. The original single-base resolution bisulfite sequencing data are shown in *Figure 4—figure supplement 1A*. Values are the mean of three independent biological replicates; error bars indicate SEM. Asterisks indicate a statistically significant difference according to Student's *t*-test. ***: p<0.001. The values of cytosine methylation in each biological replicate are shown in *Supplementary file 1*. (**B**) Percentage of methylated cytosines in the intergenic region (IR) of TYLCV in local infection assays with the V2 null mutant TYLCV (TYLCV-V2null) in *AGO4*-silenced (TRV-*NbAGO4*) or control (TRV-EV) *N. benthamiana* plants at 4 dpi, as detected by bisulfite sequencing. Samples come from the same plants used in *Figure 3E*. The original single-base resolution bisulfite sequencing data are shown in *Figure 4—figure supplement 1B*. Values are the mean of four independent biological replicates; error bars indicate SEM. Asterisks indicate a statistically significant difference according to Student's *t*-test. *: p<0.05. The values of cytosine methylation in each biological replicate are shown in *Supplementary file 1*. (**C**) Percentage of methylated cytosines in the intergenic region (IR) of TYLCV in systemic infection assays with TYLCV wild-type or V2 null mutant (TYLCV-V2null) in *AGO4*-silenced (TRV-NbAGO4) or control (TRV-EV) *N. benthamiana* plants at 3 weeks post-inoculation (wpi), as detected by bisulfite sequencing. Samples come from the same plants used in *Figure 3F*. The original single-base resolution bisulfite sequencing data are shown in *Figure 4—figure supplement 2*. Values are the mean of four independent biological replicates; error bars indicate SEM. Asterisks indicate a statistically significant difference according to Student's *t*-test. *: p<0.05. The values of cytosine methylation in each biological replicate are shown in *Supplementary file 1*.

The online version of this article includes the following figure supplement(s) for figure 4:

**Figure supplement 1.** Original single-base resolution bisulfite sequencing data of the intergenic region (IR) of TYLCV (wild-type and V2 null mutant) in local infection assays, related to *Figure 4A,B*.

**Figure supplement 2.** Original single-base resolution bisulfite sequencing data of the intergenic region (IR) of TYLCV and TYLCV-V2null in systemic infection assays, related to *Figure 4C*.

3xFLAG-NbAGO4-1 co-expressed with wild-type or V2 null mutant TYLCV in local infection assays in *N. benthamiana*, and visualized AGO4-bound vsiRNA by sRNA northern blotting. While infected samples contained both 21- and 24-nt vsiRNA, and the occurrence and accumulation of these sRNA species was not affected by the presence of virus-encoded V2, mostly 24-nt vsiRNA co-immunoprecipitated with AGO4 (*Figure 5C*). Interestingly, a higher amount of vsiRNA associated to AGO4 in the samples infected with the wild-type virus (*Figure 5C*). Taken together, these results demonstrate that V2 does not affect the production or accumulation of vsiRNA, nor does it hamper loading of the

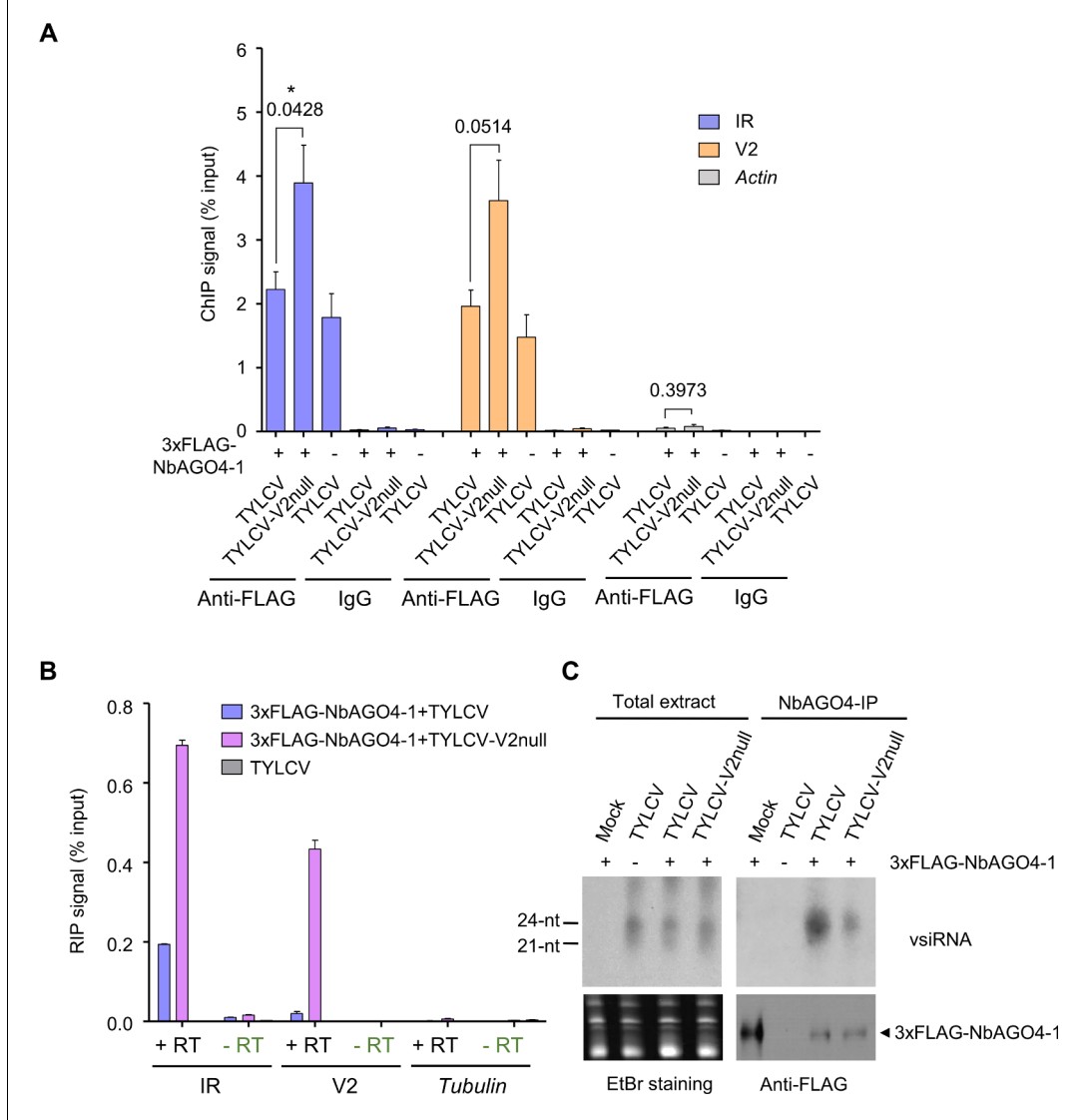

**Figure 5.** V2 interferes with AGO4 binding to the viral genome through hindering its association to the viral RNA but does not hamper production or loading of vsiRNA. (**A**) 3xFLAG-NbAGO4-1 binds the viral (TYLCV) genome. Binding was detected by chromatin immunoprecipitation (ChIP) upon transient expression in *N. benthamiana* followed by qPCR. Two regions of the viral genome, the IR and the V2 ORF, were analyzed; *Actin* was used as negative control. Values represent the mean of four independent biological replicates; error bars represent SEM. Asterisks indicate a significant difference according to Student's *t*-test; the *P*-value for the different comparisons is shown. *: p<0.05. (**B**) RNA immunoprecipitation (RIP)-based detection of viral RNA bound by NbAGO4-1 upon transient expression in *N. benthamiana* leaves infected by TYLCV or TYLCV-V2 null. *Tubulin* serves as an unbound loading control. Samples without reverse transcriptase (-RT) are used as control for DNA contamination. Values represent average qRT-PCR signal normalized to inputs of three technical replicates; error bars represent SEM. This experiment was repeated three times with similar results; additional independent biological replicates can be found in *Figure 5—figure supplement 1*. (**C**) NbAGO4-1 binds viral small interfering RNA (vsiRNA) independently of V2. Northern blot of vsiRNA in total extracts or 3xFLAG-NbAGO4-1 immunoprecipitates (NbAGO4-IP) of *N. benthamiana* leaf patches infiltrated with TYLCV wild-type or V2 null mutant infectious clones (TYLCV, TYLCV-V2null) at 2 days after infiltration. Detection was performed with a [32]P-labeled DNA probe for the intergenic region (IR).

The online version of this article includes the following figure supplement(s) for figure 5:

**Figure supplement 1.** V2 interferes with the NbAGO4-1 binding to the viral RNA.

vsiRNA molecules into AGO4, but it interferes with binding of this protein to the viral RNA and genome in order to suppress DNA methylation and promote virulence.

## Methylation of the viral genome and its suppression by V2 occur in a Cajal body-dependent manner

In Arabidopsis, AGO4 has been shown to co-localize with its interactor NRPE1 (NRPD1b), a subunit of Pol V, in the Cajal body, which was then suggested to be a center for the assembly of AGO4/NRPE1/siRNA complexes, enabling RdDM at target loci (*Li et al., 2008*; *Li et al., 2006*). However, the functional significance of this subnuclear localization has so far remained elusive. Interestingly, both V2-GFP and the different RFP-AGO4 orthologues from *N. benthamiana* and tomato co-localize in a distinct subnuclear compartment, identified as the Cajal body by the accumulation of the nucleolus and Cajal body marker fibrillarin (*Barneche et al., 2000*), upon transient expression in *N. benthamiana* (*Figure 6A*). Of note, most of nuclear V2-GFP accumulates in the Cajal body, although some fluorescence can be detected in the nucleoplasm. All AGO4 orthologues are distributed throughout the nucleoplasm and absent from the nucleolus; clear Cajal body localization can be detected for NbAGO4-1, NbAGO4-2, SlAGO4a, and SlAGO4b, while Cajal body localization of SlA-GO4d is less conspicuous (*Figure 6A*). Analysis of the V2/AGO4 interaction by bimolecular fluorescence complementation (BiFC), which is based on visualization and hence provides spatial information, unveiled that, strikingly, the association between these two proteins occurs mostly or exclusively in the Cajal body, where V2 homotypic interactions also occur (*Figure 6B*).

In order to evaluate the relevance of the Cajal body localization of the V2-AGO4 interaction, we took advantage of the fact that a GFP-V2 fusion protein, as opposed to the previously mentioned V2-GFP, does not localize to the Cajal body (*Figure 7A,B*; *Figure 7—videos 1* and *2*), otherwise showing an indistinguishable subcellular distribution pattern. Importantly, GFP-V2 still interacts with AGO4 in co-IP assays (*Figure 7—figure supplement 1A*). Nevertheless, only the Cajal body-localized V2-GFP, but not GFP-V2, can complement a null mutation in V2 in terms of suppression of methylation of the viral genome (*Figure 7C*; *Figure 7—figure supplement 1B and C*), indicating that the Cajal body localization of the V2-AGO4 interaction is essential for V2 to exert its function.

Since our results indicate that the suppression of the AGO4-mediated methylation of the viral genome occurs in a Cajal body-dependent manner, we decided to test if the Cajal body is required for this antiviral response. For this purpose, we eliminated Cajal bodies in the plant by silencing of their signature component coilin through VIGS; the effect of knocking down *coilin* on the formation of Cajal bodies has been previously demonstrated (*Shaw et al., 2014*). As expected, silencing *coilin* (*Figure 7D*) resulted in the apparent disappearance of Cajal bodies (*Figure 7—figure supplement 2A*). *Coilin*-silenced plants were then inoculated with TYLCV wild-type or V2 null mutant, and viral accumulation and methylation of the IR of the viral genome were evaluated at 3 weeks post-inoculation (*Figure 7—figure supplement 2B,C*; *Figure 7E*). A tendency toward higher viral accumulation in *coilin*-silenced plants was observed, but not statistically significant (*Figure 7—figure supplement 2C*). Strikingly, however, methylation of the IR of the viral genome in the V2 mutant virus was largely decreased by silencing of *coilin* (*Figure 7E*; *Figure 7—figure supplement 2D*; *Supplementary file 1*), suggesting that efficient methylation of the viral genome requires intact Cajal bodies. Notably, the described V2 interactor HDA6 does not localize to the Cajal body (*Figure 7—figure supplement 3*) and the V2-HDA6 complexes seem to localize outside of the nucleus (*Wang et al., 2018*), supporting the idea that the detected Cajal body-dependent effects rely on AGO4 and not HDA6.

## Discussion

The plant DNA viruses geminiviruses and pararetroviruses are both targets and suppressors of DNA methylation; this possibly extends to the third family of plant DNA viruses, nanoviruses, although experimental evidence is lacking (reviewed in *Pooggin, 2013*; *Pumplin and Voinnet, 2013*). The independent evolution of viral suppressors of DNA methylation argues for an antiviral effect of this epigenetic modification. Indeed, seminal experiments by *Brough et al., 1992* and *Ermak et al., 1993* demonstrated that methylation of the geminivirus genome interferes with its replication in transformed protoplasts, likely due to a dual effect on viral gene expression and function of the replication complex. More specifically, RdDM seems to play a prominent role in plant defence against geminiviruses, since RdDM mutants or silenced plants display increased susceptibility to geminivirus infection (*Raja et al., 2008*; *Zhong et al., 2017*), and DNA methylation and repressive histone marks typical of RdDM are deposited on the viral genome (*Castillo-González et al., 2015*; *Ceniceros-Ojeda et al., 2016*; *Coursey et al., 2018*; *Dogar, 2006*; *Jackel et al., 2016*; *Kushwaha et al.,*

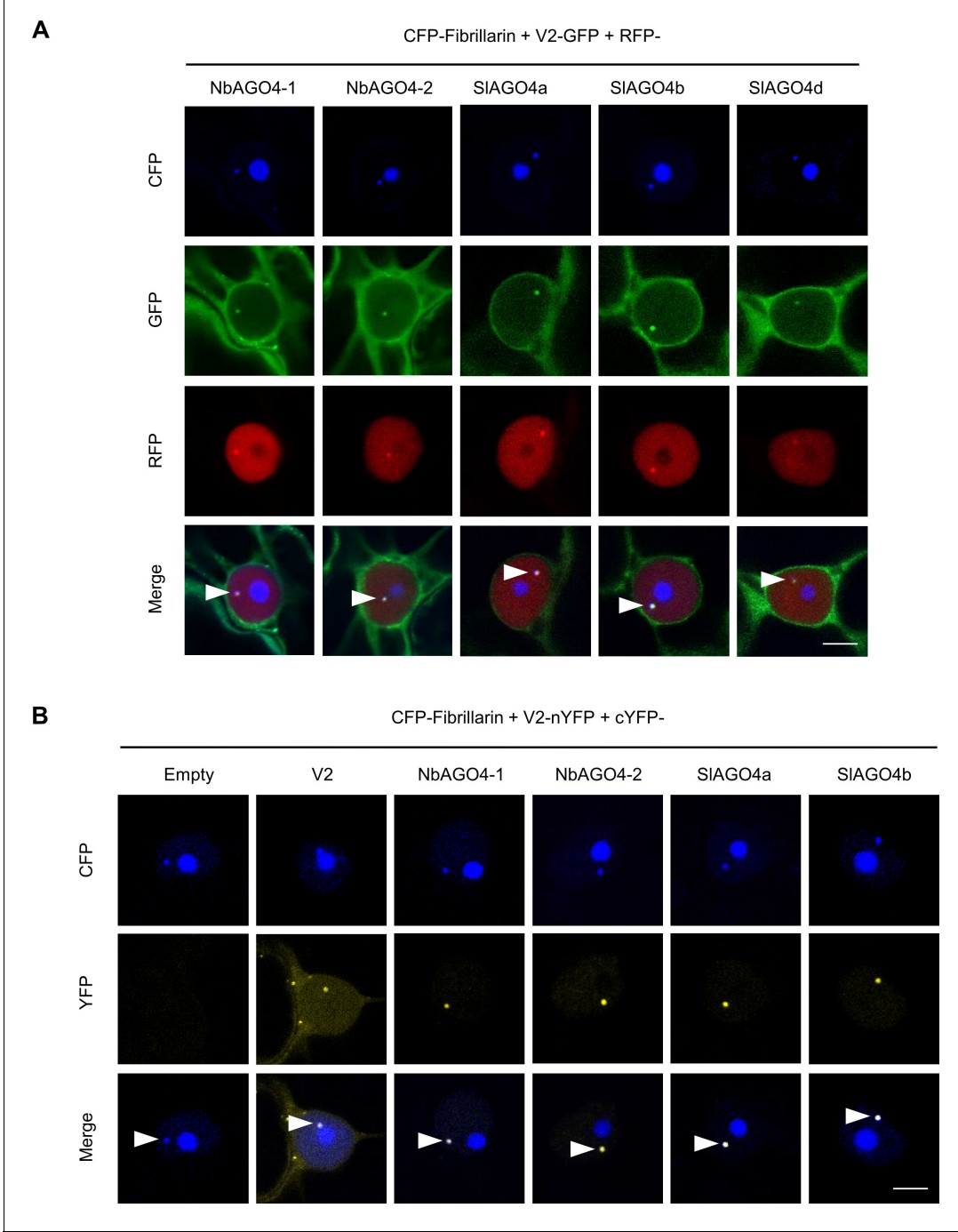

**Figure 6.** V2 interacts with AGO4 in the Cajal body. (**A**) V2-GFP and RFP-AGO4 co-localize in the Cajal body. CFP-Fibrillarin, V2-GFP and RFP-NbAGO4-1/2 or RFP-SlAGO4a/b/d were transiently co-expressed in *N. benthamiana* epidermal cells. CFP-Fibrillarin is used as nucleolus and Cajal body marker. Confocal images were taken at two days after infiltration. Arrowheads indicate the position of the Cajal body. Bar, 5 μm. This experiment was repeated more than three times with similar results. (**B**) V2 interacts with AGO4 in the Cajal body. The N-terminal half of the YFP fused to V2 (V2-nYFP) was transiently co-expressed with the C-terminal half of the YFP alone (cYFP, as negative control), or cYFP-NbAGO4, cYFP-SlAGO4, or cYFP-V2 (as positive control) in *N. benthamiana* leaves. CFP-Fibrillarin was used as nucleolus and Cajal body marker. Confocal images were taken at two days after infiltration. Yellow fluorescence indicates a positive interaction. Arrowheads indicate the position of the Cajal body. Bar, 5 μm. This experiment was repeated more than three times with similar results.

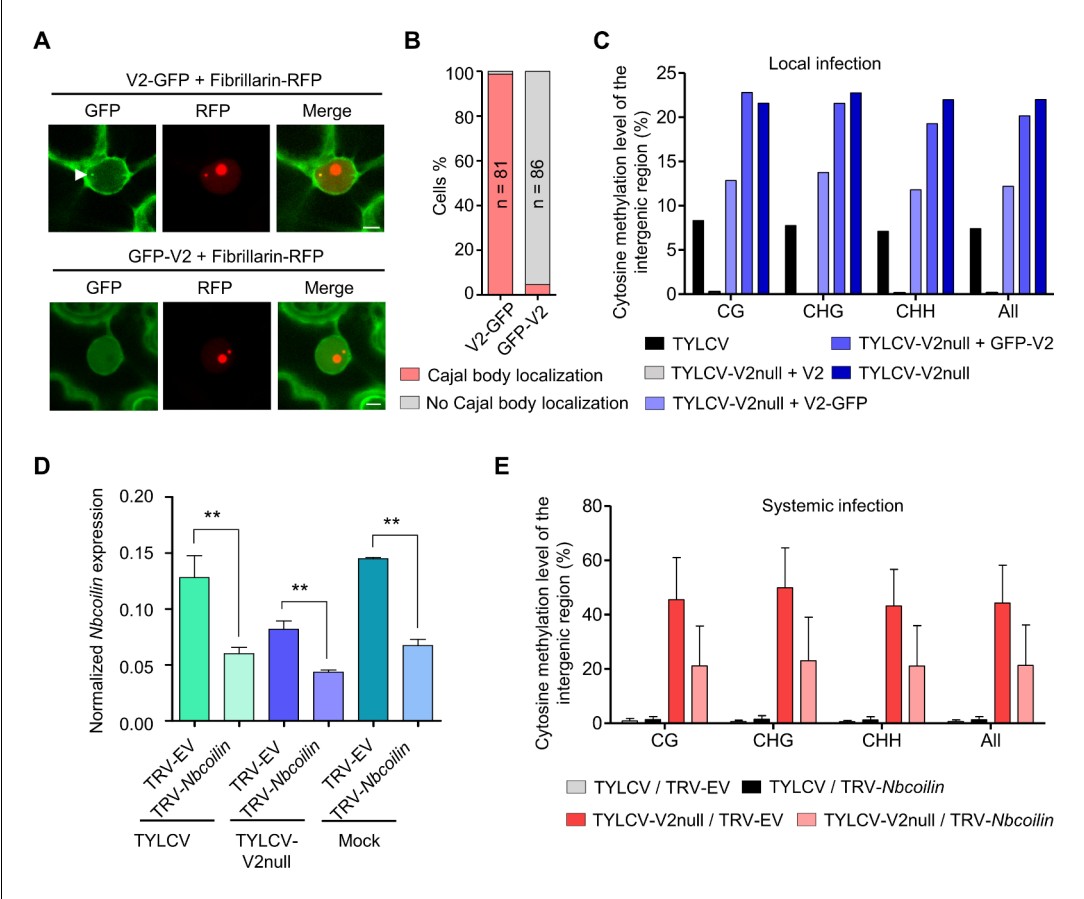

**Figure 7.** Methylation of the viral DNA and its suppression by V2 occur in a Cajal body-dependent manner. (**A**) V2-GFP co-localizes with Fibrillarin-RFP (nucleolus and Cajal body marker) in the Cajal body, while GFP-V2 does not. RFP-Fibrillarin and V2-GFP or GFP-V2 were transiently co-expressed in *N. benthamiana* epidermal cells. Confocal images were taken at two days after infiltration. Arrowheads indicate the position of the Cajal body. Bar, 5 μm. This experiment was repeated three times with similar results. (**B**) Quantification of Cajal body localization of V2-GFP or GFP-V2. (**C**) V2-GFP, but not GFP-V2, can restore the suppression of viral DNA methylation of a V2 null TYLCV mutant in local infection assays. Cytosine methylation in the intergenic region (IR) of the V2 null mutant genome in locally infected leaf patches of *N. benthamiana* expressing V2, V2-GFP, or GFP-V2 was detected by bisulfite sequencing at 3 days post-inoculation (dpi). TYLCV or TYLCV-V2null alone were used as controls. The original single-base resolution bisulfite sequencing data are shown in *Figure 7—figure supplement 1B*. This experiment was repeated twice with similar results. (**D**) *Nbcoilin* expression in *Nbcoilin*-silenced (TRV-*Nbcoilin*) and control plants (TRV-EV) infected with TYLCV, TYLCV-V2null, or mock-inoculated at 3 wpi measured by RT-qPCR. Gene expression was normalized to *NbTubulin*. Values are the mean of six independent biological replicates; error bars indicate SEM. Asterisks indicate a statistically significant difference according to Student's *t*-test. **: p<0.01. (**E**) Percentage of methylated cytosines in the intergenic region (IR) of TYLCV in systemic infection assays with TYLCV wild-type or V2 null mutant (TYLCV-V2null) in *Nbcoilin*-silenced (TRV-*Nbcoilin*) or control (TRV-EV) *N. benthamiana* plants at 3 weeks post-inoculation (wpi), as detected by bisulfite sequencing. Values are the mean of three independent biological replicates; error bars indicate SEM. The original single-base resolution bisulfite sequencing data are shown in *Figure 7—figure supplement 2D*. The online version of this article includes the following video and figure supplement(s) for figure 7:

**Figure supplement 1.** Relevance of the Cajal body localization of the V2-AGO4 interaction.

**Figure supplement 2.** TYLCV infection in *Nbcoilin*-silenced plants.

**Figure supplement 3.** NbHDA6-GFP does not localize to the Cajal body.

**Figure supplement 4.** TYLCV V2_{L76S} interacts with AGO4 in the Cajal body.

**Figure supplement 5.** Model for the V2-mediated inhibition of the AGO4-dependent methylation of the viral DNA.

**Figure 7—video 1.** Z-stack video showing the co-localization of V2-GFP and Fibrillarin-RFP in the Cajal body.

https://elifesciences.org/articles/55542#fig7video1

**Figure 7—video 2.** Z-stack video showing the localization of GFP-V2 and Fibrillarin-RFP.

https://elifesciences.org/articles/55542#fig7video2

*2017*; *Wang et al., 2018*). However, the subnuclear distribution of these antiviral methylation events is currently unknown.

AGO4 is a central component of the canonical RdDM pathway, and as such an obvious target for viral inhibition. However, AGO4 also affects susceptibility to RNA viruses and viroids, and is targeted by proteins encoded by RNA viruses, which raises the idea that either RdDM on the host genome plays a role in modulating plant-virus interactions broadly, or AGO4 has an antiviral role beyond RdDM (*Brosseau et al., 2016*; *Ma et al., 2015*; *Minoia et al., 2014*). Supporting the latter, the AGO4-dependent defence against a potexvirus is independent of other RdDM components and does not require nuclear localization of AGO4 (*Brosseau et al., 2016*).

The geminivirus TYLCV encodes the essential, multifunctional V2 protein, which acts as a suppressor of DNA methylation (*Wang et al., 2014*). Here, we show that V2 binds to the plant AGO4 in the Cajal body, and suppresses the AGO4-dependent methylation of the viral genome, which requires its Cajal body localization (*Figures 1*, *2* and *4–6*), and the AGO4-mediated restriction of viral accumulation (*Figure 3E,F*). V2 impairs binding of AGO4 to the viral RNA and to the viral genome (*Figure 5*); consequently, we hypothesize that the viral protein might mask a surface required for the complementarity-based pairing to the nascent Pol V transcript. Our results indicate that AGO4-dependent methylation of viral DNA occurs quickly in the absence of V2 (*Figure 4A,B*). Nevertheless, *AGO4* silencing still has a detectable, if minor, positive impact on the accumulation of the wild-type virus, which correlates with decreased viral DNA methylation (*Figure 3E,F*; *Figure 4C*), suggesting that the V2-mediated suppression of AGO4 function is not complete. On the other hand, wild type-like levels of viral DNA methylation are not restored in the V2 mutant upon *AGO4* silencing, which raises the idea that AGO4 might not be the only methylation-related target of V2. In agreement with this, V2 has been shown to bind HDA6 and interfere with its promotion of MET1-dependent methylation of the viral DNA (*Wang et al., 2018*).

Recently, V2 encoded by the geminivirus *Cotton leaf curl Multan virus* (CLCuMV) was found to interact with AGO4 in *N. benthamiana*, leading to enhanced viral accumulation and a reduction in viral DNA methylation (*Wang et al., 2019*). It should be noted that the V2 proteins encoded by CLCuMV and TYLCV are only 65% identical (*Figure 7—figure supplement 4A*); moreover, mutation of a conserved residue, L76, abolishes the interaction between CLCuMV V2 and NbAGO4 (*Wang et al., 2019*), but does not affect the interaction between TYLCV V2 and NbAGO4 or SlAGO4, which still occurs in the Cajal body (*Figure 7—figure supplement 4*). This mutation, however, negatively affects V2 self-interaction in the Cajal body (*Figure 7—figure supplement 4D*). These results suggest that different geminivirus species might have convergently evolved to target AGO4, underscoring the potential relevance of this host factor for the viral infection.

The finding that the physical association between TYLCV V2 and AGO4 takes places in a specific nuclear body, the Cajal body, and has an impact on the methyl-state of the viral population in the cell, suggests that either all or most viral DNA molecules must localize in this subnuclear compartment at some point of the viral cycle, or that passage of AGO4 through the Cajal body is required for its function on the methylation of the viral DNA. Supporting this notion, the activity of V2 as a suppressor of methylation of the viral genome requires the Cajal body localization of its interaction with AGO4 (*Figure 7A–C*). This observation hints at a potential functional role of the Cajal body during the viral infection; whether such a role might be linked to gene expression, DNA replication, or some other process remains to be investigated. Importantly, our results indicate that intact Cajal bodies are required for the efficient methylation of the viral genome (*Figure 7D,E*; *Figure 7—figure supplement 2*), indicating that antiviral DNA methylation indeed occurs in a Cajal body-dependent manner. In agreement with this idea, depletion of Cajal bodies through knock-down of *coilin* was previously found to promote the infection by the geminivirus *Tomato golden mosaic virus* (*Shaw et al., 2014*). The Cajal body has also been connected to systemic infection of plant RNA viruses, and proteins encoded by RNA viruses can bind coilin, which impacts plant-virus interactions (*Kim et al., 2007a*; *Kim et al., 2007b*; *Semashko et al., 2012*; *Shaw et al., 2014*; reviewed in *Ding and Lozano-Durán, 2020*), although these effects are likely independent of DNA methylation.

Based on our results, we propose a scenario in which antiviral methylation of invading geminivirus DNA in plants takes place in a Cajal body-dependent manner. In the context of the arms race between host and virus, TYLCV has evolved V2, which is required during the infection to interfere with AGO4 binding to the viral genome, suppressing methylation of the viral DNA and promoting virulence (*Figure 7—figure supplement 5*). In summary, taken together, our findings draw a long

sought-after functional connection between RdDM, the Cajal body, and antiviral DNA methylation, paving the way to a deeper understanding of DNA methylation and antiviral defence strategies in plants.

# Materials and methods

## Key resources table

| Reagent type (species) or resource | Designation | Source or reference | Identifiers | Additional information |
|---|---|---|---|---|
| Gene (*Nicotiana benthamiana*) | *NbAGO4-1* | GenBank | DQ321490 | |
| Gene (*Nicotiana benthamiana*) | *NbAGO4-2* | GenBank | DQ321491 | |
| Gene (*Solanum lycopersicum*) | *SlAGO4a* | Sol Genomics | SOLYC01G008960 | |
| Gene (*Solanum lycopersicum*) | *SlAGO4b* | Sol Genomics | SOLYC06G073540 | |
| Gene (*Solanum lycopersicum*) | *SlAGO4d* | Sol Genomics | SOLYC01G096750 | |
| Antibody | anti-V2 (Rabbit polyclonal) | This work | N/A | WB (1:2000) |
| Antibody | anti-Actin (Rabbit polyclonal) | Agrisera | Cat# AS13 2640 | WB (1:5000) |
| Antibody | anti-FLAG (Mouse monoclonal) | Sigma | Cat# F3165 | IP: (5 µg per gram of tissue) WB (1:5000) |
| Antibody | anti-HA (12CA5) (Mouse monoclonal) | Roche | Cat# 11583816001 | WB (1:5000) |
| Antibody | anti-GFP (Mouse monoclonal) | Abiocode | Cat# M0802-3a | WB (1:5000) |
| Commercial assay or kit | ClonExpress MultiS One Step Cloning Kit | Vazyme | Cat# C113-01 | |
| Commercial assay or kit | pENTR/D-TOPO Cloning Kit | Invitrogen | Cat# K240020SP | |
| Commercial assay or kit | Gateway LR Clonase II Enzyme Mix | Invitrogen | Cat# 11791100 | |
| Commercial assay or kit | QuickChange Lightning Site-Directed Mutagenesis Kit | Agilent Technologies | Cat# 210518 | |
| Commercial assay or kit | Plant RNA kit | OMEGA Bio-tek | Cat# R6827 | |
| Commercial assay or kit | iScriptTM cDNA Synthesis Kit | Bio-Rad | Cat# 1708890 | |
| Commercial assay or kit | DNeasy Plant Mini Kit | QIAGEN | Cat# 69104 | |
| Commercial assay or kit | EpiTect Plus DNA Bisulfite Kit | QIAGEN | Cat# 59124 | |
| Commercial assay or kit | QIAquick PCR Purification Kit | QIAGEN | Cat# 28106 | |
| Commercial assay or kit | Superscript III first-strand synthesis system | Thermo Fisher Scientific | Cat# 18080051 | |
| Software, algorithm | Cytosine methylation analysis Kismeth | *Gruntman et al., 2008* | PMID:18786255 | http://katahdin.mssm.edu/kismeth/revpage.pl |

## Plasmids and cloning

All primers and plasmids used for cloning are summarized in *Supplementary files 3* and *4*, respectively. To generate binary vectors to express *AGO4* from *N. benthamiana* and tomato (cv. Money maker), the full-length coding sequence of *AGO4* genes was amplified using cDNA as template. *NbAGO4-1*, *NbAGO4-2*, *SlAGO4a*, *SlAGO4b*, *SlAGO4d*, and *SlWRKY75* were cloned into pENTR/D-TOPO (Invitrogen) following the manufacturer's instructions. The binary plasmids to express RFP and 3xHA N-terminal fusions were generated by Gateway cloning the AGO4 coding sequence into pGWB555 and pGWB515, respectively (*Nakagawa et al., 2007*). V2-GFP, GFP-V2 and the TYLCV infectious clone have been previously described (*Rosas-Diaz et al., 2018*; *Wang et al., 2017a*). The V2 null TYLCV mutant was generated with the QuickChange Lightning Site-Directed Mutagenesis Kit (Agilent Technologies, Cat #210518) using the wild-type infectious clone as template. For the plasmids used for biomolecular fluorescence complementation (BiFC), *NbAGO4*, *SlAGO4*, V2/V2$_{L76S}$ were cloned into entry vector pDONR221-P1P4/pDONR221-P3P2 (Invitrogen) and then Gateway-cloned into the pBiFC-2in1-CN vector (*Grefen and Blatt, 2012*) as shown in *Supplementary file 4*. The binary plasmids for split-luciferase complementation imaging assay were generated by Gateway cloning the *NbAGO4*, *SlAGO4*, *SlWRKY75* or V2 from pENTR/D-TOPO into pGWB-N-luc and pGWB-C-luc (*Yu et al., 2019*). To generate pCAMBIA1300-3xFLAG-NbAGO4-1, pCAMBIA1300 was digested with *Xba*I and *Xma*I, and then the 3xFLAG and NbAGO4-1 coding sequences were amplified by PCR and Infusion-cloned into pCAMBIA1300 with ClonExpress MultiS One Step Cloning Kit (Vazyme). pDONR207-*Fibrillarin* (*Kim et al., 2007b*) was used to Gateway-clone *Fibrillarin* to pGWB545 (*Nakagawa et al., 2007*). NbHDA6-GFP is described in *Wang et al., 2018*.

## Plant materials and growth conditions

*N. benthamiana* and tomato plants (cv. Money maker) were grown in a controlled growth chamber under long day conditions (LD, 16 hr light/8 hr dark) at 25℃.

## *Agrobacterium*-mediated transient gene expression in *N. benthamiana*

All binary plasmids were transformed into *Agrobacterium tumefaciens* strain GV3101, with the exception of pBINTRA6, which was transformed into *A. tumefaciens* strain C58c1. *A. tumefaciens* clones carrying the constructs of interest were liquid-cultured in LB with appropriate antibiotics at 28℃ overnight. Bacterial cultures were then centrifuged at 4000 g for 10 min and resuspended in the infiltration buffer (10 mM MgCl$_2$, 10 mM MES pH 5.6, 150 μM acetosyringone) and adjusted to an OD$_{600}$ = 0.5. Next, the bacterial suspensions were incubated in the buffer at room temperature and in the dark for 2–4 hr and then infiltrated into 3- to 4-week-old *N. benthamiana* plants. For co-expression experiments, the different *Agrobacterium* suspensions were mixed at 1:1 ratio before infiltration.

## Protein extraction and immunoprecipitation assays

Protein extraction and co-immunoprecipitation assays were performed as described in *Wang et al., 2017a*. V2 and actin were detected by western blot with anti-V2 (custom-made by ABclonal with V2$_{1-116aa}$) and anti-Actin (Agrisera AS13 2640) antibodies, respectively.

Protein extracts were immunoprecipitated with GFP-Trap beads (Chromotek, Germany), and analyzed by western blot with anti-GFP (Abiocode, M0802-3a) and anti-HA (12CA5) (Roche, Cat. No. 11 583 816 001) antibodies.

For 3xFLAG-NbAGO4-1 IP followed by vsiRNA extraction, 6 grams of *N. benthamiana* leaves transiently expressing 3xFLAG-NbAGO4-1 were collected, ground in liquid nitrogen and homogenized in 6x (w:v) extraction buffer (20 mM Tris HCl pH7.5, 25 mM MgCl$_2$, 300 mM NaCl, 5 mM DTT, 0.5% NP-40, 1x complete Protease Inhibitor Cocktail (Roche)) at 4℃ with rotation for 30 min. The extract was subjected to centrifugation (14,000 rpm, 25 min) at 4℃. 5 μg anti-FLAG antibody (Sigma, F3165) per gram of tissue were added to the supernatant in a new tube and incubated at 4℃ overnight. The next day, 20 μl of slurry Protein G beads (Invitrogen) per gram of tissue were added and subjected to a further incubation for 2 hr with rotation at 4℃. After incubation, Protein G beads were washed three times in 3x (v:v) homogenate wash buffer (20 mM Tris pH7.5, 25 mM MgCl$_2$, 300 mM NaCl, 0.5% NP-40). The quality of purification was examined by SDS-PAGE followed by immunoblotting.

## Split-luciferase complementation imaging assay

Split-luciferase complementation imaging assays were performed as described (*Chen et al., 2008*). Equal volumes of *A. tumefaciens* harboring V2-N-luc or C-luc-NbAGO4-1/2, C-luc-SlAGO4a/b, or C-luc-SlWRKY75 at $OD_{600}$ = 0.5 were mixed at 1:1 ratio. Three different combinations of *A. tumefaciens* were infiltrated on the same *N. benthamiana* leaf. 1 mM luciferin (in $H_2O$) was infiltrated into the inoculated leaves 2 days after *Agrobacterium* infiltration. A low-light cooled CCD imaging apparatus (NightShade LB985 with IndiGO software) was used to capture and analyze the luciferase signal at 2 dpi.

## Confocal imaging

Confocal imaging for co-localization of V2-GFP, RFP-AGO4, and CFP-Fibrillarin upon transient expression in *N. benthamiana* epidermal cells was performed on a Leica TCS SP8 point scanning confocal microscope using the pre-set sequential scan settings for GFP (Ex:488 nm, Em:500–550 nm), RFP (Ex:561 nm, Em:600–650 nm), and CFP (Ex:442 nm, Em:452–482 nm). Confocal imaging for co-localization of Fibrillarin-RFP (*Kim et al., 2007b*) or V2-RFP and NbHDA6-GFP, or Fibrillarin-RFP and V2-GFP or GFP-V2, was performed in the same way.

## Bimolecular fluorescence complementation

For bimolecular fluorescence complementation (BiFC) analyses, *A. tumefaciens* clones carrying pBiFC-2in1-CN binary constructs (*Grefen and Blatt, 2012*) and CFP-Fibrillarin were mixed at 1:1 ratio and infiltrated into 3- to 4-week-old *N. benthamiana* plants. Imaging was performed 2 days later under a Leica TCS SP8 confocal microscope by using the pre-set sequential scan settings for YFP (Ex: 514 nm, Em: 525–575 nm) and for CFP (Ex:442 nm, Em:452–500 nm).

## Virus-induced gene silencing

The vectors used for virus-induced gene silencing (VIGS) were pBINTRA6 (*Ratcliff et al., 2001*) and pTRV2-GW (*Taylor et al., 2012*). A 362 bp fragment of *NbAGO4-1* cDNA (from nt 1920 to 2281) and a 389 bp fragment of *Nbcoilin* cDNA (from nt 2222 to 2610) were amplified using specific primers shown in *Supplementary file 3*, cloned into pENTR/D-TOPO (Invitrogen), and subcloned into pTRV2-GW through an LR reaction (Invitrogen) to yield TRV-*NbAGO4* and TRV-*Nbcoilin*. VIGS assays were performed as described in *Lozano-Durán et al., 2011*. For TYLCV local infection assays, *A. tumefaciens* carrying pBINTRA6 and TRV-*NbAGO4* or TRV-EV were mixed and inoculated into 18-day-old *N. benthamiana* plants. Two weeks later, fully expanded young leaves were infiltrated with *A. tumefaciens* carrying the TYLCV infectious clone and samples were collected at 4 days post-inoculation (dpi) to detect viral accumulation. For TYLCV systemic infection assays, *A. tumefaciens* carrying pBINTRA6 and TRV-*NbAGO4* or TRV-EV (empty vector) and the TYLCV infectious clone were mixed and inoculated into 18-day-old *N. benthamiana* plants. The three most apical leaves of each plant were collected at 3 weeks post-inoculation (wpi) to detect viral accumulation. TYLCV systemic infection assays in *Nbcoilin*-silenced plants were performed as previously described for *NbAGO4*-silenced plants.

## Quantitative PCR (qPCR) and reverse transcription PCR (RT-qPCR)

To determine viral accumulation, total DNA was extracted from *N. benthamiana* leaves (from infiltrated leaves in local infection assays and from apical leaves in systemic infection assays) using the CTAB method (*Minas et al., 2011*). Quantitative PCR (qPCR) was performed with primers to amplify Rep (*Wang et al., 2017b*). As internal reference for DNA detection, the 25S ribosomal DNA interspacer (ITS) was used (*Mason et al., 2008*). To detect *NbAGO4-1*, *NbAGO4-2*, *Nbcoilin*, and *NbAGO6* transcripts, total RNA was extracted from *N. benthamiana* leaves by using Plant RNA kit (OMEGA Bio-tek # R6827). RNA was reverse-transcribed into cDNA by using the iScriptTM cDNA Synthesis Kit (Bio-Rad #1708890) according to the manufacturer's instructions. *NbTubulin* was used as reference gene (*Liu et al., 2012*). Relative expression was calculated by the comparative Ct method (2-ΔΔCt). qPCR and RT-qPCR were performed in a BioRad CFX96 real-time system as described previously (*Wang et al., 2017b*). Total RNA was extracted from the leaves of tomato plants mock-inoculated or infected with TYLCV at 3 weeks post-inoculation (wpi). *SlActin* was used as reference gene (*Expósito-Rodríguez et al., 2008*). Similarly, RT-qPCR was performed on RNA

extracted from tomato to detect the expression of *SlAGO4a/d/c/d*. All primers used for qPCR and qRT-PCR are listed in *Supplementary file 5*.

## DNA bisulfite sequencing analysis

DNA from virus-infected plant tissues was extracted by DNeasy Plant Mini Kit (QIAGEN, Cat. No. 69104), and 500 ng of purified DNA was subjected to bisulfite treatment using EpiTect Plus DNA Bisulfite Kit (QIAGEN, Cat. No. 59124) according to the manufacturer's handbook. The selected fragment (viral IR) of the bisulfite-treated DNA was amplified by PCR (Fw: TTTGATGTATTTTTTA TTTGTTGGGGTTT, Rv: CCCTTACAACARATATAARATCCCT); amplified fragments were cloned into the pMD18-T vector by TA ligation and sequenced (>15 clones per experiment). Cytosine methylation analysis was performed with Kismeth (http://katahdin.mssm.edu/kismeth/revpage.pl) (*Gruntman et al., 2008*). Values obtained in all independent biological replicates are shown in *Supplementary file 1*; please note that despite the instrinsic variation in these experiments, the same trends in relative values (compared to control samples) consistently emerge, supporting the reliability of the results.

## RNA immunoprecipitation

The *Agrobacterium* clone carrying the binary vector to express 3xFLAG-NbAGO4-1 was co-infiltrated with those carrying the TYLCV or TYLCV-V2null infectious constructs in *N. benthamiana* leaves, and tissues were collected at 2 dpi. RNA immunoprecipitation (RIP) assays were performed as previously described (*Köster and Staiger, 2014*). In brief, 0.6 g of leaves without crosslinking were used as the input samples, while 1.5 g of leaves were subjected to crosslinking and used as for immunoprecipitation (IP). For IP samples, leaves were fixed for 15 min in 1% formaldehyde in 1xPBS buffer under vacuum; crosslinking was terminated by adding 125 mM glycine for 5 min. Then the tissue was ground to powder and homogenized in 5 ml of lysis buffer (50 mM Tris-HCl (pH 7.6), 150 mM NaCl, 25 mM MgCl$_2$, 5 mM DTT, 0.5% NP-40, 5 mM EDTA, 2 mM PMSF, 8 unit/ml Ribolock RNase inhibitor (Fisher Scientific, FEREO0384), and 1% Protease Inhibitor Cocktail (Sigma)). The samples were then centrifuged at 16,000 g for 20 min. The extract was subjected to immunoprecipitation with FLAG antibody (Sigma, F3165) bound to Dynabeads Protein G (Invitrogen) for 2 hr with gentle rotation at 4°C. After incubation, beads were washed three times in lysis buffer.

Immunocomplexes were eluted with 200 µl of Elution buffer (1% SDS, 0.1 M NaHCO$_3$) at 60°C for 15 min. After reverse crosslinking, 4 µl EDTA 0.5 M, 8 µl Tris and 40 µg proteinase K (Invitrogen) were added to each sample, which was incubated at 42°C for 1 hr.

Input and immunoprecipitated RNA were extracted with TRIzol reagent (Thermo Fisher Scientific) and resuspended in 200 µl or 20 µl ddH$_2$O, respectively. 20 µl RNA from IP samples and 5 µl RNA from input samples were treated with Turbo DNase (Fisher Scientific, NC9075048) at 37°C for 1 hr and then reverse-transcribed with Superscript III first-strand synthesis system (Thermo Fisher Scientific) with random hexamers. qPCR was used to determine the relative enrichment for each sample, which was calculated by normalizing the value of immunoprecipitated RNA to that of the input.

## Chromatin immunoprecipitation (ChIP) assay

The *Agrobacterium* clone carrying the binary vector to express 3xFLAG-NbAGO4-1 was co-infiltrated with those carrying the TYLCV or TYLCV-V2null infectious clones in *N. benthamiana* leaves, and tissues were collected at 2 dpi. Chromatin immunoprecipitation (ChIP) assays were performed as described (*He et al., 2018*). In brief, the cross-linking of 2 g of leaves was performed with 1% formaldehyde in 1xPBS buffer and stopped with 1/15 vol of 2 M glycine by vacuum infiltration. Then the tissue was ground to powder and resuspended in HB buffer (2.5% Ficoll 400, 5% Dextran T40, 0.4 M Sucrose, 25 mM Tris pH 7.4, 10 mM MgCl2, 0.035% β-mercaptoethanol, 1% Protease Inhibitor Cocktail (Sigma)), homogenized and filtered through Miracloth (Milli-pore). Triton x-100 was added to the supernatant to a final concentration of 0.5%. After spinning at 2000 g for 20 min at 4°C, the pellet was re-suspended in HB buffer containing 0.1% Triton x-100 and spun at 2000 g for 10 min at 4°C. Isolated nuclei were re-suspended in 500 µl of Nuclei Lysis buffer and sonicated by BioruptorTM UCD-200 sonicator (Diagenode) for 30 min. Following centrifugation at 21,130 g for 5 min at 4°C, the supernatant was separated and used for input and immunoprecipitation. After adding 9 vol of ChIP dilution buffer to the supernatant, this was pre-cleared with 10 µl of Dynabeads Protein G

(Invitrogen) for 1 hr at 4°C. After removing the beads from the mixture, the supernatant was incubated with anti-FLAG antibody (Sigma, F3165), or anti-IgG antibody (Sigma, I5006) overnight at 4°C. The following day, after adding 20 µl of Dynabeads Protein G, the mixture was incubated for 2 hr at 4°C. Beads were sequentially washed with 1 ml of the following buffers: Low-Salt Wash buffer (150 mM NaCl, 0.1% SDS, 1% Triton x-100, 2 mM EDTA, 20 mM Tris pH 8.0), High-Salt Wash buffer (500 mM NaCl, 0.1% SDS, 1% Triton x-100, 2 mM EDTA, 20 mM Tris pH 8.0), LiCl wash buffer (250 mM LiCl, 1% Igepal, 1% Sodium Deoxycholate, 1 mM EDTA, 10 mM Tris pH 8.0), TE buffer (10 mM Tris pH 8.0, 1 mM EDTA). Immunocomplexes were eluted with 250 µl of Elution buffer (1% SDS, 0.1 M NaHCO$_3$) at 65°C for 15 min. After reverse crosslinking, 10 µl of 0.5 M EDTA, 20 µl of 1 M Tris pH 6.5 and 1 µl of proteinase K (Invitrogen) were added to each sample, which was incubated at 45°C for 2 hr. DNA was then purified using QIAquick PCR Purification Kit (QIAGEN, Cat. No. 28106). The products were eluted into 200 µl of ddH$_2$O, and analyzed by qPCR. The primers used in this experiment are listed in *Supplementary file 5*; the primers for *Actin* are taken from *Maimbo et al., 2010*.

### Small RNA (sRNA) extraction and northern blot analysis

Small RNA (sRNA) extraction and northern blot were performed as described (*Yang et al., 2016*). Briefly, sRNAs were purified from total extracts or AGO4 immunoprecipitates and subjected to northern blot analysis. For each sample, sRNAs were separated on a 17% polyacrylamide gel, which was electrotransferred to a Hybond N+ membrane (GE Lifesciences). Membranes were cross-linked, incubated for 2 hr at 65°C, and hybridized overnight at 38°C with $^{32}$P-labeled probes for the intergenic region (IR) of the viral genome amplified by PCR (Fw: TCCTCTTTAGAGAGAGAACAA TTGGGA, Rv: ACAACGAAATCCGTGAACAG) or oligonucleotides in PerfectHyb buffer (Sigma). Washed membranes were exposed to X-ray films at −80°C for 3 days.

## Acknowledgements

The authors thank Xinyu Jian, Aurora Luque, and Yujing (Ada) Liu for technical assistance, Gang Yu, Alberto Macho, and Xueping Zhou for sharing materials, and all members in Rosa Lozano-Duran's and Alberto Macho's groups for stimulating discussions and helpful suggestions. The authors are grateful to Alberto Macho, Huang Tan, and Tamara Jimenez-Gongora for critical reading of the manuscript.

## Additional information

### Funding

| Funder | Grant reference number | Author |
|---|---|---|
| Chinese Academy of Sciences | XDB27040206 | Rosa Lozano-Duran |
| National Natural Science Foundation of China | 31671994 | Rosa Lozano-Duran |
| National Natural Science Foundation of China | 31870250 | Rosa Lozano-Duran |

The funders had no role in study design, data collection and interpretation, or the decision to submit the work for publication.

### Author contributions

Liping Wang, Conceptualization, Investigation, Writing - review and editing; Yi Ding, Li He, Guiping Zhang, Investigation, Writing - review and editing; Jian-Kang Zhu, Supervision, Funding acquisition, Writing - review and editing; Rosa Lozano-Duran, Conceptualization, Supervision, Funding acquisition, Writing - original draft

### Author ORCIDs

Liping Wang  https://orcid.org/0000-0002-0812-1566
Li He  https://orcid.org/0000-0001-8289-4076

Jian-Kang Zhu ![ORCID] https://orcid.org/0000-0001-5134-731X
Rosa Lozano-Duran ![ORCID] https://orcid.org/0000-0001-7348-8842

**Decision letter and Author response**
Decision letter https://doi.org/10.7554/eLife.55542.sa1
Author response https://doi.org/10.7554/eLife.55542.sa2

## Additional files

### Supplementary files
• Supplementary file 1. Values of IR methylation in independent experiments and biological replicates.

• Supplementary file 2. Exclusive unique peptide count of NRPE1 co-immunoprecipitated with NbAGO4-1 in the presence or absence of V2 as identified by AP-MS.

• Supplementary file 3. List of primers used for cloning in this study.

• Supplementary file 4. List of plasmids used in this study.

• Supplementary file 5. List of primers used for qPCR and qRT-PCR in this study.

• Transparent reporting form

### Data availability
All data generated during this study are included in the manuscript and supporting files.

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
