## [Decision Letter]

**Acceptance summary:**

Your study reports on a Geminivirus suppressor, V2, that counteracts AGO4-mediated methylation of viral DNA upon host plant infection through physical interaction with AGO4. AGO4 is central to the RNA-directed DNA methylation machinery that has been found to localize to so-called Cajal bodies. Your findings make Cajal bodies central hubs for antiviral defense. In addition, this work is important as it reveals that evolution of plant resistance mechanisms (here viral DNA methylation) is counteracted by the evolution of viral mechanisms suppressing host plant immunity, a phenomenon reported many times for bacterial, fungal or oomycete effectors, but less so from plant-virus interactions.

**Decision letter after peer review:**

Thank you for submitting your article "A virus-encoded protein suppresses methylation of the viral genome in the Cajal body through its interaction with AGO4" for consideration by *eLife*. Your article has been reviewed by three peer reviewers, including Thorsten Nürnberger as the Reviewing Editor and Reviewer #1, and the evaluation has been overseen by Christian Hardtke as the Senior Editor.

The reviewers have discussed the reviews with one another and the Reviewing Editor has drafted this decision to help you prepare a revised submission.

Your study provides a nice examples of how plant resistance mechanisms (here viral DNA methylation) is counteracted by the evolution of viral mechanisms suppressing host plant immunity, a phenomenon reported for bacterial, fungal or oomycete effectors, but less so from plant-virus interactions. This, together with the finding that Cajal bodies are central hubs for antiviral defense in plants is undoubtedly important for furthering our molecular mechanistic understanding of plant virus interactions.

The reviewers feel, however, that your manuscript requires essential revisions that are listed below.

Essential revisions:

1) Major aspects described in your study have been already reported from this or related viral systems. These include the immunosuppressive role of geminiviral V2 proteins on RdDM (Wang et al., 2018; 2019), V2-AGO4 interaction (Wang et al., 2019), the requirement of Cajal body for anti-geminiviral role (Shaw et al., 2014), AGO4 binding to DNA in RdDM (Lahmy, Genes and Development, 2016), the cajal body localization of AGO4 and its role in RdDM (Li et al., 2006; Li et al., 2008; Dou, Nucleic Acids Research, 2013). Please address these concerns carefully as advance over existing knowledge is an important criterion for publication in our journal.

2) In the light of the above said it appears most desirable to provide additional mechanistic insight into how V2 brings about suppression of AGO4-mediated viral DNA methylation. This could be, for example, achieved by sequential ChIP analysis. Likewise, it is important to provide unambiguous evidence that the effect reported may not be brought about by V2-HDA6 interaction. This is best achieved by providing a V2 mutant that may still bind HDA6, but is unable to bind to AGO4 and to suppress antiviral defense.

3) There is also concern about data variation in DNA methylation quantification. For example, the CG methylation percentages in Figure 4A are approximately 2% and 55% for 3dpi/TYLCV and 3dpi/TYLCV-V2null samples, respectively. In Figure 7B, the corresponding percentages are 8% and 22%. Great differences can also be found between and Figure 4C and Figure 7D (systemic infection) despite of the fact that plant materials used in Figure 4 and Figure 7 are comparable. This needs an explanation.

---

## [Author Response]

Essential revisions:1) Major aspects described in your study have been already reported from this or related viral systems. These include the immunosuppressive role of geminiviral V2 proteins on RdDM (Wang et al., 2018; 2019), V2-AGO4 interaction (Wang et al., 2019), the requirement of Cajal body for anti-geminiviral role (Shaw et al., 2014), AGO4 binding to DNA in RdDM (Lahmy, Genes and Development, 2016), the cajal body localization of AGO4 and its role in RdDM (Li et al., 2006; Li et al., 2008; Dou, Nucleic Acids Research, 2013). Please address these concerns carefully as advance over existing knowledge is an important criterion for publication in our journal.

It is indeed known that AGO4 is a central component of the RdDM pathway, and that it localizes in the Cajal body, although the connection between RdDM and this subnuclear compartment has not been further functionally dissected. The function of the geminivirus V2 protein as a TGS suppressor has in fact been previously described, and its association with AGO4 recently shown in the case of the geminivirus Cotton leaf curl Multan virus (Wang et al., 2019). Nevertheless, the localization of V2 in the Cajal body and the specificity of its interaction with a RdDM regulator at this site have never been reported; the relevance of the Cajal body for anti-viral DNA methylation is, to our knowledge, also demonstrated here for the first time; additionally, our work provides insight into the molecular mechanism by which this viral protein interferes with the function of AGO4, showing that the interaction hampers the AGO4 association to the viral genome, probably through hindering the binding to the scaffold RNA (see new Figure 5B and Figure 5—figure supplement 1). Therefore, we believe that this work not only sheds light on long sought-after connections between the Cajal body, RdDM, and anti-viral DNA methylation, but also furthers our understanding of the viral manipulation of the plant RdDM at the molecular level.

2) In the light of the above said it appears most desirable to provide additional mechanistic insight into how V2 brings about suppression of AGO4-mediated viral DNA methylation. This could be, for example, achieved by sequential ChIP analysis.

Following the reviewers advice, we have performed new experiments to try to obtain additional mechanistic insight into the suppression of AGO4 mediated by V2.

Since V2 physically interacts with AGO4 and interferes with its binding to the viral genome, our hypothesis was that V2 likely covers a surface in the AGO4 proteins required for the interaction with an ‘anchoring’ element, e.g. NRPE1 or the PolV lncRNA transcript. Following with this idea, we have undertaken a double approach: (i) We have performed IP-MS experiments of tagged AGO4 in the presence of WT or a V2 null mutant TYLCV to identify differential interacting partners; (ii) We have performed RIP experiments with tagged AGO4 in the presence of WT or a V2 null mutant TYLCV to evaluate the capacity of AGO4 to bind virus-derived RNA in the presence or absence of V2. The results of these experiments are now included in Supplementary file 2 and Figure 5B/Figure 5—figure supplement 1.

As seen in new Supplementary file 2, peptides corresponding the NRPE1 can be detected in the AGO4 immunoprecipitates, and no significant quantitative differences can be observed between samples with or without V2. Nevertheless, since V2 might be affecting only a subpopulation of the total AGO4 pool, these results do not rule out the possibility that the viral protein might be interfering with the interaction of AGO4 with NRPE1 or other components of the RdDM machinery.

The RIP experiments presented in Figure 5B/Figure 5—figure supplement 1 show a decrease in the binding of AGO4 to viral RNAs corresponding to the IR (non-coding) or the V2 coding sequence in the presence of the V2 protein. Therefore, the interaction with V2 seems to interfere with the association of AGO4 to the viral scaffold RNA, hence impairing its binding to the viral DNA and the subsequent recruitment of DRM2 and methylation of the viral genome.

Likewise, it is important to provide unambiguous evidence that the effect reported may not be brought about by V2-HDA6 interaction. This is best achieved by providing a V2 mutant that may still bind HDA6, but is unable to bind to AGO4 and to suppress antiviral defense.

We agree with the reviewers in that having a mutant V2 version enabling the uncoupling of the interactions with HDA6 and AGO4 would be an ideal tool to dissect the relative contribution of these targets to anti-viral DNA methylation; unfortunately, such a hypothetical mutant version is not currently available. Nevertheless, we have performed co-localization experiments between HDA6 and V2, which clearly demonstrate that HDA6 is not accumulated in the Cajal body (new Figure 7—figure supplement 3); the BiFC results shown in Wang et al., 2018, Figure 1B, support this finding, given that the detected interaction between HDA6 and V2 occurs outside of the nucleus. Since the suppressive effect of V2 on viral DNA methylation detected in this work requires localization of the viral protein in the Cajal body (Figure 7), this effect is most likely the result of the interaction with Cajal body-localized AGO4, and not with HDA6. It is possible that both AGO4 and HDA6 are targeted by V2 and contribute to different steps of viral DNA methylation.

3) There is also concern about data variation in DNA methylation quantification. For example, the CG methylation percentages in Figure 4A are approximately 2% and 55% for 3dpi/TYLCV and 3dpi/TYLCV-V2null samples, respectively. In Figure 7B, the corresponding percentages are 8% and 22%. Great differences can also be found between and Figure 4C and Figure 7D (systemic infection) despite of the fact that plant materials used in Figure 4 and Figure 7 are comparable. This needs an explanation.

The level of DNA methylation of the viral genome is the final result of the combined influence of a number of factors, reflecting the status of the dynamic interaction between virus and host plant at the time of sampling. Intrinsic variation can be detected between different batches of this experiments; nevertheless, all biological replicates include the appropriate controls, with which comparisons are made.

Please note that we used at least 6 independent plants for each group of samples in each biological replicate, and at least three biological replicates were performed in Figure 4C and Figure 7E. As shown in Supplementary file 1 (“Values of IR methylation in independent experiments and replicates”), although the absolute values of DNA methylation from each biological replicate are different, the same trends in relative values consistently emerge, supporting the reliability of the results.